# Hypertriglyceridaemia-Induced Acute Pancreatitis: A Different Disease Phenotype

**DOI:** 10.3390/diagnostics12040868

**Published:** 2022-03-31

**Authors:** Greta Dancu, Felix Bende, Mirela Danila, Roxana Sirli, Alina Popescu, Cristi Tarta

**Affiliations:** 1Center for Advanced Research in Gastroenterology and Hepatology, Department of Internal Medicine II, Division of Gastroenterology and Hepatology, “Victor Babes” University of Medicine and Pharmacy Timisoara, Eftimie Murgu Sq. No. 2, 300041 Timisoara, Romania; gretadancu@yahoo.com (G.D.); bendefelix@gmail.com (F.B.); mireladanila@gmail.com (M.D.); roxanasirli@gmail.com (R.S.); 2Department X, 2nd Surgical Clinic, Researching Future Chirurgie 2, “Victor Babes” University of Medicine and Pharmacy Timisoara, Eftimie Murgu Sq. No. 2, 300041 Timisoara, Romania; tarta.cristi@umft.ro

**Keywords:** acute pancreatitis, hypertriglyceridaemia, severity prediction, inflammation markers

## Abstract

Acute pancreatitis (AP) is the most common gastrointestinal indication requiring hospitalisation. Severe hypertriglyceridaemia (HTG) is the third most common aetiology of AP (HTGAP), with a complication rate and severity that are higher than those of other aetiologies (non-HTGAP). The aim of this study was to evaluate the supposedly higher complication rate of HTGAP compared to non-HTGAP. The secondary objectives were to find different biomarkers for predicting a severe form. This was a retrospective study that included patients admitted with AP in a tertiary department of gastroenterology and hepatology. The patients were divided into two groups: HTGAP and non-HTGAP. We searched for differences regarding age, gender, the presence of diabetes mellitus (DM), the severity of the disease, the types of complications and predictive biomarkers for severity, hospital stay and mortality. A total of 262 patients were included, and 11% (30/262) of the patients had HTGAP. The mean ages were 44.4 ± 9.2 in the HTGAP group and 58.2 ± 17.1 in the non-HTGAP group, *p* < 0.0001. Male gender was predominant in both groups, at 76% (23/30) in the HTGAP group vs. 54% (126/232) in non-HTGAP, *p* = 0.02; 53% (16/30) presented with DM vs. 18% (42/232), *p* < 0.0001. The patients with HTG presented higher CRP 48 h after admission: 207 mg/dL ± 3 mg/dL vs. non-HTGAP 103 mg/dL ± 107 mg/dL, *p* < 0.0001. Among the patients with HTGAP, there were 60% (18/30) with moderately severe forms vs. 30% (71/232), *p* = 0.001, and 16% (5/30) SAP vs. 11% (27/232) in non-HTGAP, *p* = 0.4 Among the predictive markers, only haematocrit (HT) and blood urea nitrogen (BUN) had AUCs > 0.8. According to a multiple regression analysis, only BUN 48 h was independently associated with the development of SAP (*p* = 0.05). Diabetes mellitus increased the risk of developing severe acute pancreatitis (OR: 1.3; 95% CI: 0.1963–9.7682; *p* = 0.7). In our cohort, HTGAP more frequently had local complications compared with non-HTGAP. A more severe inflammatory syndrome seemed to be associated with this aetiology; the best predictive markers for complicated forms of HTGAP were BUN 48 h and HT 48 h.

## 1. Introduction

Acute pancreatitis (AP) is a common worldwide cause of morbidity and mortality, occurring in 40 out of 100,000 people in the Western population [1,2], and it is the most common gastrointestinal condition requiring hospitalisation [3]. Recent studies reported that the prevalence of inpatient hospitalisations for AP significantly increased in the preceding decade [4,5].

Severe hypertriglyceridaemia (HTG), defined as triglyceride (TG) ≥1000 mg/dL, is a well-described risk factor for AP. It is the third most common aetiology after excessive alcohol intake and gallstone disease and is the most common aetiological factor (up to 56%) for AP in pregnant women [6,7]. HTG accounts for 7–10% of AP cases [8].

The most accepted theory of pathophysiological mechanisms is that excess TG is hydrolysed by pancreatic lipase, resulting in free fatty acids (FFAs). The excess FFAs consequently overwhelm the binding capacity of albumin and cause acinar and pancreatic capillary injury. In addition, hyperviscosity resulting from chylomicronaemia causes impaired pancreatic blood flow, leading to ischaemia and acidosis, ultimately resulting in further pancreatic injury [9,10].

Patients with HTGAP had higher rates of complications and more severe forms [11,12]; they more frequently required stays in the intensive care unit (ICU) [13,14] due to either severe forms of AP or the need for double-filtration plasmapheresis (DFPP) to reduce the circulating TG concentration.

As HTG is associated with diabetes mellitus (DM), some studies found that the risk of HTGAP was higher in diabetic patients [15] and that patients affected by HTGAP and DM were younger [16]. The published data suggest that scoring systems for AP should also take the aetiology into consideration [17]. However, there is a paucity of data regarding their usefulness in HTGAP patients. Newer studies explored more simple predictive markers of AP severity, such as blood urea nitrogen (BUN) and haematocrit (HT) or the neutrophil-to-lymphocyte ratio (NLR) and platelet-to-lymphocyte ratio (PLR) [18,19,20].

The main objective of this study was to compare the severity of HTGAP with that of AP of other aetiologies. The secondary objectives were to compare different severity-predicting markers and scores for severe forms of AP based upon aetiologies: HTGAP versus other aetiologies (non-HTGAP), such as the BUN, haematocrit (HT), neutrophil-to-lymphocyte ratio (NLR), platelet-to-lymphocyte ratio (PLR), and bedside index for severity in AP (BISAP).

## 2. Materials and Methods

### 2.1. Patient Selection

We retrospectively reviewed the electronic medical records of patients who were admitted with AP in a tertiary department of gastroenterology from 1 January 2018 to 31 August 2019. We identified potential patients using the discharge diagnoses of AP (i.e., ICD-9 code (577.0) and ICD-10 code (K85.9)). We confirmed the diagnosis of AP based on the revised Atlanta criteria guidelines, which require at least two of the following three criteria to be fulfilled: (1) abdominal pain consistent with acute pancreatitis (the acute onset of a persistent, severe, epigastric pain often radiating to the back); (2) serum lipase activity (or amylase activity) at least three times greater than the upper limit of normal and (3) characteristic findings of acute pancreatitis on contrast-enhanced computed tomography (CECT) and, less commonly, magnetic resonance imaging (MRI) or transabdominal ultrasonography [21]. The exclusion criteria were pregnant women, patients younger than 18 years, missing data, and patients with recurrent pancreatitis or cases defined as more than one attack of AP that had completely or almost completely resolved within 3 months.

### 2.2. Stratification Based on Severity

According to the revised Atlanta criteria, AP was divided into three degrees of severity: mild (MAP), moderately severe (MSAP) and severe (SAP). MAP was characterised by the absence of organ failure and the absence of local or systemic complications; MSAP included the presence of transient organ failure and/or local complications, whereas SAP was defined by the presence of persistent organ failure (≥48 h) [21]. The local complications included ascites, pleural effusion, pancreatic necrosis, abdominal fluid collections, pseudocysts and thrombosis of the spleno-portal axis; the distant complications included organ and system failure, which included respiratory, cardiovascular and renal systems. Organ failure was defined as a score of two or more for one of these three organ systems according to the modified Marshall scoring system [22]. Transient organ failure was considered to be failure that resolved within 48 h, whereas persistent organ failure lasted beyond 48 h.

### 2.3. Patients Characteristics

The following information was collected and recorded from the patients’ charts: gender, age, blood pressure (mmHg), respiratory rate (breaths per minute), oxygen saturation (%), pulse rate (heart beats per minute), body mass index (BMI), BISAP score at admission, leucocyte number at admission (cells number/μL), thrombocyte number at admission (cells number/μL), haematocrit at admission and at 48 h (%), glucose level at admission (mg/dL), aspartate aminotransferase (AST) at admission (U/L), alanine aminotransferase (ALT) at admission (U/L), total bilirubin (TB) at admission (mg/dL), gamma-glutamyl transpeptidase (GGT) at admission (U/L), alkaline phosphatase (FAL) at admission (U/L), creatinine level at admission and at 48 h (mg/dL), BUN at admission and at 48 h (mg/dL), CRP at 48 h (mg/dL), NLR at admission and at 48 h, PLR at admission and at 48 h, the aetiology of AP, complications and mortality. 

HTGAP was diagnosed when TG levels were ≥1000 or ≥500 mg/dL and milky serum and no other aetiology was found. Based on the aetiology, the patients were split in two groups: HTGAP and non-HTGAP.

BMI was calculated as weight in kg/(height in m)^2^.

Based on the recorded data, NLR and PLR were calculated at admission and at 48 h, using the formulas: NLR = neutrophil count/lymphocyte count and PLR = platelet count/lymphocyte count.

The BISAP score was evaluated at admission using the worst parameters available in the first 24 h. The five-point BISAP score system incorporates the following variables: BUN level >25 mg/dL, an impaired mental status, systemic inflammatory response syndrome (SIRS), age >60 years and the presence of pleural effusion. One point was assigned for each variable within 24 h of presentation and added for a composite score of 0–5 [23].

The patients were followed up for 90 days after discharge, with visits to the outpatient clinic or by phone.

### 2.4. Ethical Approval

The Emergency County Timisoara Hospital Committee for Ethics approved this study (decision number 206, 7 September 2020).

### 2.5. Statistical Analysis

The variables are expressed as the mean ± SD or median (range) and categorical data, as percentages, as appropriate. Univariate analysis was used to calculate *p* values and odds ratios (ORs). Multivariate logistic regression analyses were used to assess whether the inflammation markers were independent factors for predicting MSAP + SAP or SAP in patients with AP. The accuracy of each marker for predicting SAP was assessed using receiver operating characteristic (ROC) curves, and the analysis was carried out using the MedCalc 15.0 software. The positive likelihood ratio (+LR) and negative likelihood ratio (−LR) were also calculated using the following formulas: +LR = sensitivity/(100 − specificity); −LR = (100 − sensitivity)/specificity. A 2-tailed value of *p* < 0.05 was considered statistically significant.

## 3. Results

### 3.1. Patients’ Characteristics 

During the designated study period, a total of 282 patients with AP were identified in the records and retrospectively analysed. Meeting the exclusion criteria, 20 patients were excluded: 4 pregnant women, 2 patients under 18 years old, 8 patients with recurrent AP and 6 patients with missing data (see Figure 1). There were 30 patients diagnosed with HTGAP, representing 11%, and 232 with non-HTGAP, representing 88% (of which 53% (140/262) were biliary, 16% (43/262) were alcohol induced, in 14% (39/262) the aetiology was unknown, 1% (4/262) had pancreatic adenocarcinoma and 2% (6/262) had other aetiologies). The median age was 57 ± 17 years old, and there were 113 women (43%) and 149 men (56%).

The patients’ characteristics are presented in Table 1.

A total of 141 (53%) individuals were diagnosed as MAP, 89 (33%) presented as MSAP and 32 (12%) as SAP. The forms of AP of different severity of the two groups are also presented in Table 1. 

We analysed the types of complications of the two groups, and there were significant differences between the rates of local complications in HTGAP versus non-HTGAP. 

Regarding comorbidities, a significant difference was only observed in the number of patients with DM, the number was higher in HTG patients (*p* < 0.0001).

There were no significant differences regarding the length of hospitalisation between the two groups: the average lengths of hospital stay were 8 ± 6.7 days for the HTGAP group and 8 ± 9.3 days for the non-HTGAP group (*p* = 1). No significant differences were observed regarding ICU admission (*p* = 0.8) or the need for surgery (*p* = 0.8).

There was a single death in the HTGAP group, representing 3%, and 22 cases of death in the non-HTGAP group, representing 9% (*p* = 0.2).

A total of three patients with HTG (10%) underwent DFPP after admission; the rest of the patients were treated with oral fenofibrate. 

### 3.2. Severity Prediction

We performed an ROC analysis for parameters that are known to reflect severity, to evaluate their strength in predicting SAP and MSAP + SAP in HTG and non-HTG patients. The obtained cut-off values, ROC, sensitivity (SE), specificity (SP), *p* values, +LR and −LR are summarised in Table 2, Table 3, Table 4 and Table 5 and Figure 2 and Figure 3.

In multiple regression analysis, out of the parameters with the best AUROC, only BUN 48 h was independently associated with SAP in HTGAP. In the HTGAP group, the presence of DM increased the risk of developing SAP (OR: 1.3; 95% CI: 0.1963–9.7682; *p* = 0.7).

There were differences between predictors of severity among the two groups. For predicting SAP, the markers that had a good AUROC for the HTGAP group were HT 48 h, BUN 0 h and BUN 48 h, for non-HTGAP, they were CRP 48 h and NLR 48 h. When we combined MSAP and SAP forms, the predictors with a good AUROC were HT 48 h for HTGAP and none for non-HTGAP (Table 6).

## 4. Discussion

In our cohort, HTG was the third most frequent aetiology in AP, representing 11% of all the cases after biliary AP and alcohol-induced AP. A recent study demonstrated a mean prevalence of 14%, with a wide range of prevalence estimates (8–31%), depending on the region of the globe where the study was performed and the ethnicity of the patients included [2]. Given its high prevalence, HTG should always be considered in patients with non-biliary and non-alcohol-induced AP.

According to our data, the patients in the HTG group were significantly younger; they were predominantly males, and DM was frequently associated (53%). Higher percentages of patients (70.9%) have been found in other studies with a previous diagnosis of DM [10]. The prevalence of type 2 diabetes is rising, accompanied by a reduction in its average age at onset. Thus, diabetes, formerly a disorder of middle or old age, is increasingly found in adolescents and young adults [24]. Diabetic dyslipidaemia (DD) among patients with type 2 diabetes is very common (a prevalence of 72–85%). The main quantitative lipoprotein abnormalities of DD are increased TGs and reduced HDL cholesterol [25].

A history of DM and younger age were found to be independent risk factors for AP in patients with severe HTG, thereby increasing the ICU admission rate and prolonging hospital stays [16]. In our study, the presence of DM was associated with more severe forms of AP in the group of HTGAP; however, the difference did not reach statistical significance in our study. On the other hand, there are studies, with larger cohorts of patients that show statistically significant differences in the case of associated DM [26].

Lipid mediators are widely appreciated for their important roles in proinflammatory processes. TG metabolism is associated with inflammation and insulin resistance; moreover, even if TGs are not inherently toxic to the pancreas, their breakdown into FFAs by pancreatic lipase causes lipotoxicity. The severity of pancreatitis is dependent on the severity of the inflammatory response and the injury caused by lipotoxicity. FFAs increase inflammatory mediators such as TNF-alpha, interleukin-6 and interleukin-10, and these inflammatory cytokines play an important role in HTGAP. In obese mice and humans, hypertrophied adipocytes have been found to secrete large amounts of monocyte chemoattractant protein-1 (MCP-1), a chemoattractant that enhances macrophage infiltration into adipose tissues and the production of cytokines [27]. This may be an explanation as to why different studies showed more severe clinical findings in patients with HTGAP compared to those with AP due to other causes, e.g., higher serum levels of CRP, higher CT severity indices, more peripancreatic fluid collection and necrosis, higher severity according to the revised Atlanta classification and longer hospital stays [28]. Plasmapheresis used in the treatment of HTGAP has been shown not only to lower TG levels but also to reduce the levels of proinflammatory factors, thus contributing to the alleviation of the disease severity [29]. There studies showing that visceral fat might be one factor for worse prognosis in AP, patients with HTG usually have metabolic syndrome, which is related to an increased visceral fat volume [30].

HTGAP patients had higher levels of CRP (*p* < 0.0001) and rarely had uncomplicated forms, and MAP was observed in only 23% of cases compared to 57% in AP due to other aetiologies. Most patients with HTGAP had MSAP according to the revised Atlanta classification (*p* = 0.0005). Similar observations were made in a French cohort [31] and in three large Chinese studies [2].

Although complicated forms were frequent in the HTGAP group, the mortality rate was not higher than in the non-HTGAP group, probably due to the younger age in the HTGAP group and the equal share of severe forms, there were also no significant differences in the comorbidities, such as cardiovascular diseases or chronic renal failure, that were observed in the two studied groups, which may have contributed to more severe forms. There are literature data that show consistent findings regarding mortality [26].

There has been a constant search for rapid biomarkers for adequate prognosis prediction for AP. The markers of systemic inflammation are determined using readily available laboratory tests, including the NLR, PLR, HT, BUN and CRP. Elevation of the NLR during the first 48 hours of admission was significantly associated with SAP and was an independent negative prognostic indicator in AP [32]. Multiple studies have reported a significant association between the serum levels of BUN, HT and creatinine on admission and their changes over 24–48 h with severe disease [33,34]. According to Li et al., BISAP may be the best criterion for predicting the severity and prognosis of HTGAP [12].

Wang et al. concluded that NLR represents an inexpensive, readily available test with promise for predicting the disease severity in HTGAP with a cut-off value of 10 and AUROC of 0.8905 [20]. However, in our study, its usefulness was not demonstrated, nor was that of most of the analysed biomarkers, possibly due to the low number of MAP cases in the HTGAP group. In our study, the BUN at 48 h had the best predictive value for SAP, followed by the BUN at presentation, both with +LR of 10; the BUN at presentation was revealed in the literature as a possible marker of aggravating HTGAP, but this study lacked the analysis of BUN at 48 h [19]. Higher HT at 48 h and CRP at 48 h were associated with only a slight increase in the probability of developing SAP, according to the +LR calculated. In order to evaluate the strength of the biomarkers for predicting the overall complication rate for HTGAP, we combined MSAP and SAP. HT 48 h had an AUROC of 0.85, with LH+ of 3.8 and a *p* = 0.005 for predicting the forms with complications.

The main limitation of this study was that it was a single-centre retrospective study. The number of patients with uncomplicated forms of HTGAP was low, with no apparent causes; one reason might be that we are a referral centre for the western region of our country, and it was predominantly the more severe cases of HTGAP that were referred to our hospital. Due to the same situation, another possible source of bias was the high percentage of cases with local or general complications, MSAP and SAP, representing 46% of the total cases.

## 5. Conclusions

In our cohort, HTGAP more frequently had local complications compared with non-HTGAP and, consecutively, more severe forms of AP, statistically significant only for MSAP. A more severe inflammatory syndrome seemed to be associated with this aetiology. The best predictive markers for complicated forms of HTGAP were BUN 48 h and HT 48 h.

## Figures and Tables

**Figure 1 diagnostics-12-00868-f001:**
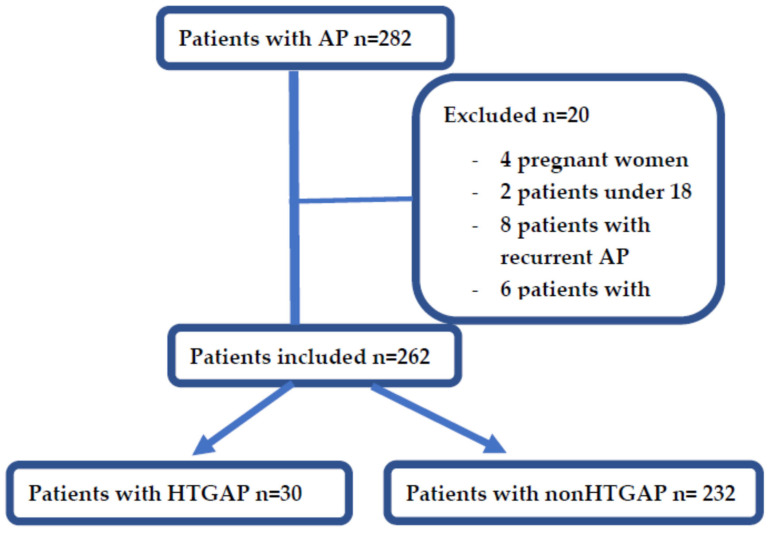
Flowchart of patients’ selection.

**Figure 2 diagnostics-12-00868-f002:**
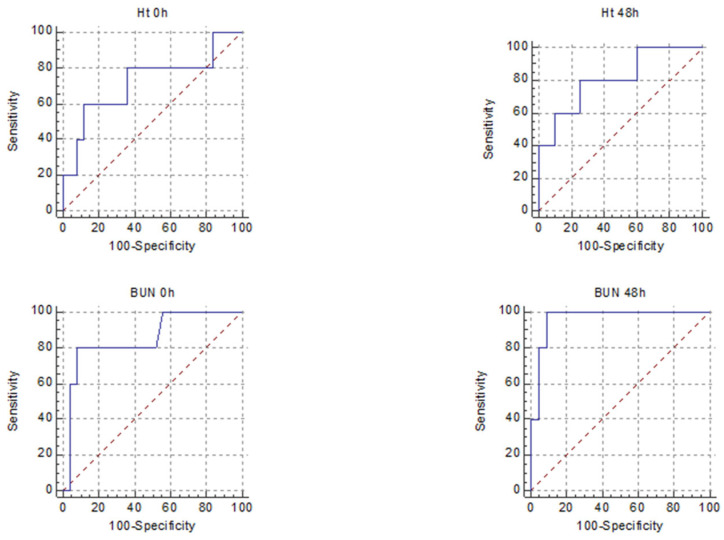
Receiver operating characteristic curves for the five best predictors of SAP in HTGAP (HT 0 h, HT 48 h, BUN 0 h, BUN 48 h, BISAP).

**Figure 3 diagnostics-12-00868-f003:**
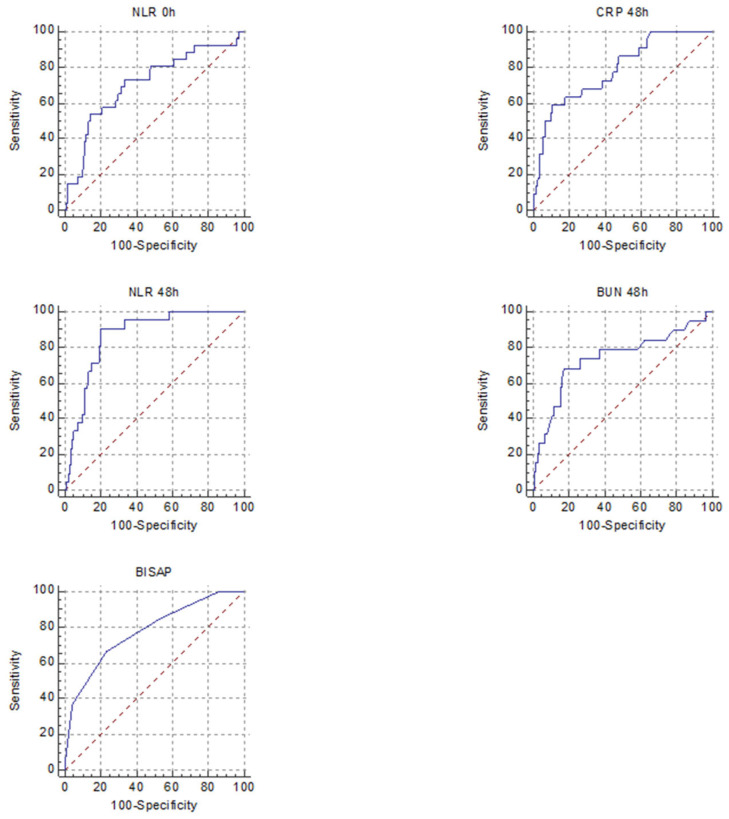
Receiver operating characteristic curves for the five best predictors of SAP in non-HTGAP (CRP 48 h, NLR 0 h, NLR 48 h, BUN 48 h, BISAP).

**Table 1 diagnostics-12-00868-t001:** Patients’ characteristics.

Patients’ Characteristics	HTGAP	Non-HTGAP	
Mean age	44.4 ± 9.2	58.2 ± 17.1	*p* < 0.0001
Male gender	76% (23/30)	54% (126/232)	*p* = 0.02
**Severity forms**			
MAP	23% (7/30)	57% (134/232)	*p* = 0.0005
MSAP	60% (18/30)	30% (71/232)	*p* = 0.001
SAP	16% (5/30)	11% (27/232)	*p* = 0.4
**Type of complications**			
Local complications	60% (18/30)	27% (64/232)	*p* = 0.0005
Organ failure	16% (5/30)	14% (34/232)	*p* = 0.9
**Laboratory data**	**Median (range)**	**Median (range)**	
Glucose level 0 h	175 (87–540)	131 (32–486)	*p* = 0.007
ALT 0 h	41 (10–315)	98 (7–1963)	*p* = 0.0002
AST 0 h	30(11–657)	104 (11–1747)	*p* = 0.0002
TB 0 h	0.8 (0.4–4.8)	1.3 (0.2–15)	*p* = 0.002
FAL 0 h	76 (38–178)	159 (33–781)	*p* = 0.0001
GGT 0 h	87 (23–1853)	212 (19–4072)	*p* = 0.004
Creatinine 0 h	0.8 (0.4–4.6)	1 (0.4–7.1)	*p* = 0.9
Creatinine 48 h	0.8 (0.3–7.4)	0.8 (0.4–7.9)	*p* = 0.6
Leucocyte 0 h	12,950 (4200–24,300)	11,300 (1300–29,630)	*p* = 0.5
Thrombocyte 0 h	210,000 (101,000–358,000)	205,000 (43,000–555,000)	*p* = 0.6
Lipase 0 h	2490 (398–21,467)	4816 (43–275,291)	*p* = 0.2
Triglyceride 0 h	1874 (512–21,110)	187 (85–351)	*p* < 0.0001
CRP 48 h	243 (8–411)	71 (1–606)	*p* < 0.0001
NLR 0 h	5.8 (2–13)	8.1 (1–17)	*p* = 0.01
NLR 48 h	7.2 (0.9–13)	5 (0.8–30)	*p* = 0.1
PLR 0 h	115 (64–279)	186 (72–383)	*p* = 0.01
PLR 48 h	131 (58–346)	169 (64–412)	*p* = 0.3
HT 0 h	43 (26–50)	39 (13–55)	*p* = 0.04
HT 48 h	38 (27–46)	36 (22–52)	*p* = 0.9
BUN 0 h	29 (9–123)	50 (8–246)	*p* = 0.001
BUN 48 h	28 (6–189)	40 (7–253)	*p* = 0.4
BISAP 0 h	2 (0–3)	2 (0–5)	*p* = 0.3
**Comorbidities**			
Diabetes mellitus	53% (16/30)	18% (42/232)	*p* < 0.0001
Arterial hypertension	10% (3/30)	16% (38/232)	*p* = 0.5
Hepatic steatosis	23% (7/30)	20 (47/232)	*p* = 0.8
Heart failure	0% (0/30)	3% (8/232)	*p* = 0.7
Chronic kidney disease	0% (0/30)	4% (10/232)	*p* = 0.5
BMI kg/m^2^	**Mean ± SD**	**Mean ± SD**	
	28.8 ± 7.6	27.4 ± 8.4	*p =* 0.8
**Length of hospital stay**	**Median (range)**	**Median (range)**	
Number of days	7.5 (3–23)	7.4 (2–42)	*p* = 0.7
**ICU admission**			
	9% (3/30)	8% (20/232)	*p* = 0.8
**Need for surgery**			
	6% (2/30)	5% (13/232)	*p* = 0.8
Mortality			
	3% (1/30)	9% (22/232)	*p* = 0.2

HTGAP = hypertriglyceridaemia-induced acute pancreatitis, MAP = mild acute pancreatitis; MSAP = moderately severe acute pancreatitis; SAP = severe acute pancreatitis, AST = aspartate aminotransferase, ALT = alanine aminotransferase, TB = total bilirubin, FAL = alkaline phosphatase, GGT = gamma-glutamyl transpeptidase, CRP = C-reactive protein, NLR = neutrophil-to-lymphocyte ratio, PLR = platelet-to-lymphocyte ratio, HT = haematocrit, BUN = blood urea nitrogen; BISAP = bedside index for severity of acute pancreatitis, BMI = body mass index, ICU = intensive care unit.

**Table 2 diagnostics-12-00868-t002:** ROC analysis of predictive factors for SAP in HTGAP.

Parameter	Cut-Off Value	AUROC	95% CI	SE	SP	+LR	−LR	
CRP 48 h	>188	0.71	0.51 to 0.86	100%	58%	2.3	0	*p* = 0.02
NLR 0 h	>4.1	0.57	0.38 to 0.75	100%	28%	1.3	0	*p* = 0.56
NLR 48 h	>5.8	0.66	0.45 to 0.83	100%	52%	2	0	*p* = 0.17
PLR 0 h	>120	0.66	0.46 to 0.82	80%	64%	2.8	0.3	*p* = 0.25
PLR 48 h	>199	0.69	0.48 to 0.85	60%	85%	4	0.4	*p* = 0.2
HT 0 h	<36	0.72	0.52 to 0.86	60%	88%	5	0.4	*p* = 0.16
HT 48 h	<35.2	0.81	0.60 to 0.93	80%	75%	3.2	0.2	*p* = 0.01
BUN 0 h	>37	0.85	0.67 to 0.95	80%	92%	10	0.2	*p* = 0.001
BUN 48 h	>45	0.96	0.80 to 0.99	100%	90%	10	0	*p* < 0.0001
BISAP	>1	0.73	0.54 to 0.87	80%	52%	1.6	0.3	*p* = 0.07

AUROC = area under the receiver operating characteristics, 95% = 95% confidence interval, SE = sensibility, SP = specificity, +LR = positive likelihood ratio, −LR = negative likelihood ratio, CRP = C-reactive protein, NLR = neutrophil-to-lymphocyte ratio, PLR = platelet-to-lymphocyte ratio, HT = haematocrit, BUN = blood urea nitrogen, BISAP = bedside index for severity of acute pancreatitis.

**Table 3 diagnostics-12-00868-t003:** Multiple regression analysis for SAP in HTGAP.

Independent Variables	Coefficient	Std. Error	Rpartial	t	
(Constant)	−0.2992				
BISAP	0.1572	0.09695	0.3570	1.621	*p* = 0.1
BUN 48 h	0.005976	0.002198	0.5396	2.719	*p* = 0.01
NLR 48 h	0.02770	0.02261	0.2774	1.225	*p* = 0.2
NLR 0 h	−0.04694	0.02699	−0.3792	−1.739	*p* = 0.09
CRP 48 h	0.0006369	0.0007024	0.2090	0.907	*p* = 0.3

CRP = C-reactive protein; NLR = neutrophil-to-lymphocyte ratio; BUN = blood urea nitrogen; BISAP = bedside index for severity of acute pancreatitis

**Table 4 diagnostics-12-00868-t004:** ROC analysis of predictive factors for MSAP + SAP in HTGAP.

Parameter	Cut-Off Value	AUROC	95% CI	SE	SP	+LR	−LR	
CRP 48 h	>147	0.58	0.38 to 0.76	72%	57%	1.7	0.4	*p* = 0.5
NLR 0 h	>9.2	0.64	0.45 to 0.81	30%	100%	0	0.7	*p* = 0.2
NLR 48 h	>5.3	0.71	0.50 to 0.87	76%	80%	3.8	0.3	*p* = 0.1
PLR 0 h	>102	0.59	0.39 to 0.76	73%	57%	1.7	0.4	*p* = 0.4
PLR 48 h	>131	0.7	0.49 to 0.86	57%	100%	0	0.4	*p* = 0.04
HT 0 h	<43	0.76	0.57 to 0.89	65%	85%	4.5	0.4	*p* = 0.005
HT 48 h	<43	0.85	0.66 to 0.96	95%	75%	3.8	0.06	*p* = 0.002
BUN 0 h	>34	0.6	0.40 to 0.77	39%	100%	0	0.6	*p* = 0.3
BUN 48 h	>35	0.57	0.36 to 0.76	40%	100%	0	0.5	*p* = 0.5
BISAP	>1	0.74	0.55 to 0.88	60%	71%	2.1	0.5	*p* = 0.02

AUROC = area under the receiver operating characteristics, 95% CI = 95% confidence interval, SE = sensibility, SP = specificity, +LR = positive likelihood ratio, −LR = negative likelihood ratio, CRP = C-reactive protein, NLR = neutrophil-to-lymphocyte ratio, PLR = platelet-to-lymphocyte ratio, HT = haematocrit, BUN = blood urea nitrogen, BISAP = bedside index for severity of acute pancreatitis, ROC = area under the curve, SE = sensibility, SP = specificity, +LR = positive likelihood ratio, −LR = negative likelihood ratio.

**Table 5 diagnostics-12-00868-t005:** ROC analysis of predictive factors for SAP in non-HTGAP.

Parameter	Cut-Off Value	AUROC	95% CI	SE	SP	LR+	LR−	
CRP 48 h	>234	0.81	0.72 to 0.84	71%	88%	5.9	0.3	*p* < 0.0001
NLR 0 h	>9.6	0.72	0.64 to 0.76	73%	66%	2.9	0.4	*p* = 0.0003
NLR 48 h	>8.1	0.83	0.80 to 0.91	90%	80%	4.5	0.1	*p* < 0.0001
PLR 0 h	>179	0.64	0.54 to 0.67	65%	62%	1.7	0.5	*p* = 0.08
PLR 48 h	>163	0.62	0.58 to 0.73	57%	70%	1.9	0.6	*p* = 0.009
Ht 0 h	<34	0.55	0.45 to 0.58	22%	81%	2.3	0.8	*p* = 0.7
Ht 48 h	<35	0.68	0.52 to 0.68	59%	74%	2.2	0.6	*p* = 0.1
BUN 0 h	>56	0.68	0.63 to 0.75	51%	86%	3.9	0.5	*p* = 0.001
BUN 48 h	>60	0.74	0.65 to 0.81	68%	82%	3.9	0.3	*p* = 0.001
BISAP	>2	0.76	0.71 to 0.82	66%	76%	2.8	0.4	*p* < 0.0001

AUROC = AUROC = area under the receiver operating characteristics, 95% CI = 95% confidence interval, SE = sensibility, SP = specificity, +LR = positive likelihood ratio, −LR = negative likelihood ratio, CRP = C-reactive protein, NLR = neutrophil-to-lymphocyte ratio, PLR = platelet-to-lymphocyte ratio, HT = haematocrit, BUN = blood urea nitrogen, BISAP = bedside index for severity of acute pancreatitis, ROC = area under the curve, SE = sensibility, SP = specificity, +LR = positive likelihood ratio, −LR = negative likelihood ratio.

**Table 6 diagnostics-12-00868-t006:** ROC analysis of predictive factors for MSAP + SAP in non-HTGAP.

Parameter	Cut-Off Value	AUROC	95% CI	SE	SP	+LR	−LR	
CRP 48 h	>93	0.7	0.71 to 0.82	67%	75%	2.7	0.4	*p* < 0.0001
NLR 0 h	>11.4	0.6	0.57 to 0.70	40%	82%	2.3	0.7	*p* = 0.0002
NLR 48 h	>6	0.7	0.68 to 0.82	71%	76%	3.9	0.3	*p* < 0.0001
PLR 0 h	>126	0.5	0.45 to 0.58	69%	40%	1.1	0.7	*p* = 0.6
PLR 48 h	>157	0.5	0.49 to 0.65	47%	69%	1.5	0.7	*p* = 0.1
HT 0 h	<44	0.5	0.45 to 0.58	36%	70%	1.2	0.8	*p* = 0.06
HT 48 h	<35	0.6	0.54 to 0.70	46%	79%	1.8	0.7	*p* = 0.007
BUN 0 h	>54	0.5	0.51 to 0.64	32%	90%	3.2	0.7	*p* = 0.03
BUN 48 h	>48	0.6	0.55 to 0.72	49%	89%	4.8	0.5	*p* = 0.004
BISAP	>2	0.7	0.64 to 0.76	50%	81%	3.9	0.5	*p* < 0.0001

AUROC = area under the receiver operating characteristics, 95% CI = 95% confidence interval, SE = sensibility, SP = specificity, +LR = positive likelihood ratio, −LR = negative likelihood ratio, CRP = C-reactive protein, NLR = neutrophil-to-lymphocyte ratio, PLR = platelet-to-lymphocyte ratio, HT = haematocrit, BUN = blood urea nitrogen, BISAP = bedside index for severity of acute pancreatitis, ROC = area under the curve, SE = sensibility, SP = specificity, +LR = positive likelihood ratio, −LR = negative likelihood ratio.

## Data Availability

Data are available on request and after hospital approval.

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
