# Peer review of "Hypertriglyceridaemia-Induced Acute Pancreatitis: A Different Disease Phenotype"

_diagnostics, 2022, doi:10.3390/diagnostics12040868_

Round 1

Reviewer 1 Report

 In this manuscript, the authors presented that, HTGAP more frequently had local complications compared with non-HTGAP and, consecutively, more severe forms of AP. I have some questions and also, I advise the authors to make some modifications in the manuscript.

Major points

#1.    According to the authors’ data, the patients in the HTG group were significantly younger; they were predominantly males, and DM was frequently associated (53%). Higher percentages of patients (70.9%) had been found with a previous diagnosis of DM. SAP and MSAP were more frequent in HTG group. Why SAP and MSAP were more in HTG group? Due to only etiology? Please describe these points in the discussion section. Also, I request the authors to show BMI of the patients. BMI may be higher in the HTG group, due to their cahracteritics. As previously reported, visceral adipose tissue may have association with severity of acute pancreatitis (Pancreatology. 2020 Sep;20(6):1056-1061.). Please discuss these points in the discussion section.

#2.    According to the authors’ opinion, the presence of DM increased the risk of developing SAP (OR: 1.3; 95% Cl: 0.1963-9.7682; p = 0.7). I think this result was not statistical significant. Please describe about this point.

#3.    According to the authors’ data, there was a single death in the HTGAP group, representing 3%, and 22 cases of death in the non-HTGAP group, representing 9% (p = 0.2). On the other hands, SAP and MSAP were more frequent in HTG group. Were these conflicting results only due to age demographic? What were the reasons of death in the non-HTGAP group? Were there any other factors? Please discuss these points in the discussion section.

Author Response

Reviewer 1

Dear Reviewer,

We highly appreciate your insightful and helpful comments on our manuscript.

We addressed all of your concerns, please see below, in red, our response to your comments.

 #1.    According to the authors’ data, the patients in the HTG group were significantly younger; they were predominantly males, and DM was frequently associated (53%). Higher percentages of patients (70.9%) had been found with a previous diagnosis of DM. SAP and MSAP were more frequent in HTG group. Why SAP and MSAP were more in HTG group? Due to only etiology? Please describe these points in the discussion section. Also, I request the authors to show BMI of the patients. BMI may be higher in the HTG group, due to their cahracteritics. As previously reported, visceral adipose tissue may have association with severity of acute pancreatitis (Pancreatology. 2020 Sep;20(6):1056-1061.). Please discuss these points in the discussion section.

We further explained the probable reasons of more severe forms of AP in HTG patients in lines 279-296. Lipid accumulation not only causes direct pancreatic toxicity, but also increases proinflammatory state, systemic inflammation being the pathophysiological mechanism responsible for the severity of AP, thus local and distant complications. We also added BMI to Table 1. comment inserted about visceral fat.

#2.    According to the authors’ opinion, the presence of DM increased the risk of developing SAP (OR: 1.3; 95% Cl: 0.1963-9.7682; p = 0.7). I think this result was not statistical significant. Please describe about this point.

You have raised an important point here, thank you. We added some observation about this point to the discussion, line 274-276.

#3.    According to the authors’ data, there was a single death in the HTGAP group, representing 3%, and 22 cases of death in the non-HTGAP group, representing 9% (p = 0.2). On the other hands, SAP and MSAP were more frequent in HTG group. Were these conflicting results only due to age demographic? What were the reasons of death in the non-HTGAP group? Were there any other factors? Please discuss these points in the discussion section.

Thank you for the suggestion, we also analyzed comorbidities and added data to Table 1 and also added some remarks to line 305-308, in the discussion.

Reviewer 2 Report

Dear Greta Dancu and the team,

I had the pleasure in reading your interesting manuscript about the hypertriglyceridemia and acute pancreatitis.  I have a few questions;

  1. Do you anticipate any long-term complications with this group of patients?
  2. Avoid acronyms without describing them in the text in the beginning.
  3. Did you notice retinopathy in this group of patients?
  4.  Did any of your patients undergo surgery for abdominal compartment syndrome?
  5. A graphical representation of the ROC analysis would be better.

Best wishes.

Author Response

Dear Reviewer,

We are thankful for your time spent reading and reviewing our paper.

Here is a point-by-point response to your comments and concerns (in red).

  1. Do you anticipate any long-term complications with this group of patients?

We have followed up our patients for 90 days after discharge, fortunately no other complications of the acute pancreatitis were observed in the HTGAP group, however, given that most of these patients are overweight, most of them present diabetes mellitus and they also present dyslipidaemia, complications of these diseases may appear, like cardiovascular disease or microangiopathy.

  1. Avoid acronyms without describing them in the text in the beginning.

We apologies for this mistake; we have included the description missing in the introduction and in tables.

  1. Did you notice retinopathy in this group of patients?

It would have been very interesting to assess this problem, however, we have no data regarding retinopathy in our database.

  1. Did any of your patients undergo surgery for abdominal compartment syndrome?

Yes, some patients needed surgery. We included data regarding surgery in Table 1. Thank you for your suggestion. we don't have the data about the reasons that led to surgery.

  1. A graphical representation of the ROC analysis would be better.

We agree. Accordingly, we added graphical representations of the ROC analysis.

Best wishes.

Reviewer 3 Report

The manuscript reports data about 30 patients with hypertriglyceridemia-induced acute pancreatitis (AP) in comparison to over 200 patients with AP of other etiology, treated in single center in the same time period. I have read the manuscript with interest, however, I was disappointed by low quality of presentation of results. Please find below my detailed suggestions regarding the revision of the manuscript.

Major issues:

The Results should be significantly revised. The findings should be ordered according to the study aims. The authors claim that “the main objective of this study was to compare the severity of HTGAP with AP of other aetiologies. The secondary objectives were to compare different severity-predicting markers and scores for severe forms of AP based upon aetiologies: HTGAP versus other aetiologies (non-HTGAP), such as the BUN, haematocrit (HT), neutrophil-to-lymphocyte ratio (NLR), platelet-to-lymphocyte ratio (PLR), and bedside index for severity in AP (BISAP).” Consequently, the Results should include, first, the comparison between HTG versus non-HTG groups, including all studied data, and then, the markers of severity in HTG versus non-HTG AP. I do not see any information about markers of AP severity in non-HTG AP – are they different comparing to HTG-induced AP?

To consistently present the differences between HTG and non-HTG groups, tables 1, 2, 3 and 7 should be merged and the resulting table should summarize all important patients’ characteristics. If possible, it should also include data on pre-existing comorbidities (obesity and metabolic syndrome should be included), the course of AP (local and systemic complications, organ failure) and the treatment of AP (e.g. the need for intensive care, surgery, time of hospital stay, mortality) in both groups. The laboratory results (CRP, NLR, PLR) should be reported with a clear information about time of sample collection (24h, 48h will be enough, but please include these for all laboratory data).

Table 4: All abbreviations should be explained. Please report 95% confidence intervals for areas under the ROC curves (please also change the heading “ROC” into “area under ROC curve” or use the abbreviation “AUROC”). Table 5: please add 95% confidence interval for AUROC.

In the text of results, the Authors state that BUN was “independently associated with the development of SAP”, which suggest multiple regression. However, Table 6 apparently shows the  results of simple (univariate) regression. If the Authors performed multiple regression analysis (which should be done), the complete data on the regression model (including all independent variables) should be presented in Table 6. Otherwise, if BUN was only analyzed in simple regression, Table 6 is not necessary, and the result should only be cited in main text, avoiding the word “independently”. Moreover, table 6 is simply copied-pasted from the statistical software, some statistics (coefficient for constant, t) are not necessary.

It would increase readability if the authors introduce the subheadings in Methods and Results.

Please be accurate in your conclusions: “In our cohort, HTGAP more frequently had local complications compared with non-HTGAP and, consecutively, more severe forms of AP.” – only MSAP was significantly more common in HTGAP in your study.

Minor issues:

The following sentences in the abstract should be rephrased to be more easily understandable: ”Male gender was predominant in the HTGAP group, at 76% (23/30) vs. 54% female 22 (126/232), p = 0.02; 53% (16/30) presented with DM vs. 18% who did not (42/232), p < 0.0001. The 23 patients with HTG presented higher CRP 48 h after admission: 207 mg/dl ± 3 mg/dl vs. 103 mg/dl ± 24 107 mg/dl, p < 0.0001.”

Line 105: I suggest to cancel the sentence “Based on the recorded data for the NLR, the PLR was calculated at admission and at 105 48 h, using the international criteria (NLR = neutrophil count/lymphocyte count; PLR = platelet count/lymphocyte count).”. One cannot calculate PLR based on NLR.

Methods: “The patients were followed up for 90 days after discharge, with visits to the outpatient clinic or by phone.” – I wonder if this was also done retrospectively? (This would be suggested by the date of Ethical Committee approval).

Statistical analysis: The Authors state in Methods that median (range) is presented for non-normally distributed variables, however, I only see means and SD in Results (while I am sure that some variables, like e.g. CRP, are non-normally distributed in the studied groups).

Line 134: “14% (39/262) had no etiology” – it is better to say that the etiology was unknown.

Figure 1. The data are fine, however, the flow chart must be arranged more aesthetically – its visual quality is very low.

Author Response

Dear Reviewer,

We would like to thank you for your time spent reviewing our paper. Your insightful comments helped to improve our paper.

Here is a point-by-point response to your comments and concerns (in red).

The Results should be significantly revised. The findings should be ordered according to the study aims. The authors claim that “the main objective of this study was to compare the severity of HTGAP with AP of other aetiologies. The secondary objectives were to compare different severity-predicting markers and scores for severe forms of AP based upon aetiologies: HTGAP versus other aetiologies (non-HTGAP), such as the BUN, haematocrit (HT), neutrophil-to-lymphocyte ratio (NLR), platelet-to-lymphocyte ratio (PLR), and bedside index for severity in AP (BISAP).” Consequently, the Results should include, first, the comparison between HTG versus non-HTG groups, including all studied data, and then, the markers of severity in HTG versus non-HTG AP. I do not see any information about markers of AP severity in non-HTG AP – are they different comparing to HTG-induced AP?

Thank you for pointing this out. We agree, therefore, we have replaced Table 1, 2, 3 and 7 with one single table (Table1), that includes information regarding patients’ characteristics, the severity of the disease, type of complication, laboratory parameters (we included all parameters studied), comorbidities, information regarding ICU admission, the need for surgery, length of hospital stay and mortality. We also included a data regarding the predicting value of the selected parameters in nonHTGAP patients (Table4,5).

To consistently present the differences between HTG and non-HTG groups, tables 1, 2, 3 and 7 should be merged and the resulting table should summarize all important patients’ characteristics. If possible, it should also include data on pre-existing comorbidities (obesity and metabolic syndrome should be included), the course of AP (local and systemic complications, organ failure) and the treatment of AP (e.g. the need for intensive care, surgery, time of hospital stay, mortality) in both groups. The laboratory results (CRP, NLR, PLR) should be reported with a clear information about time of sample collection (24h, 48h will be enough, but please include these for all laboratory data).

As we mentioned earlier, we summarized all of the information in one, more consistent table (Table1).

Table 4: All abbreviations should be explained. Please report 95% confidence intervals for areas under the ROC curves (please also change the heading “ROC” into “area under ROC curve” or use the abbreviation “AUROC”). Table 5: please add 95% confidence interval for AUROC.

Thank you for the observation, we explained all the abbreviations in all of the tables, we corrected ROC for AUROC, and we added 95% confidence intervals.

In the text of results, the Authors state that BUN was “independently associated with the development of SAP”, which suggest multiple regression. However, Table 6 apparently shows the  results of simple (univariate) regression. If the Authors performed multiple regression analysis (which should be done), the complete data on the regression model (including all independent variables) should be presented in Table 6. Otherwise, if BUN was only analyzed in simple regression, Table 6 is not necessary, and the result should only be cited in main text, avoiding the word “independently”. Moreover, table 6 is simply copied-pasted from the statistical software, some statistics (coefficient for constant, t) are not necessary.

Thank you for your suggestion. Accordingly, we added Table 3., containing all independent variables used in the multiple regression analysis.

It would increase readability if the authors introduce the subheadings in Methods and Results.

we added subheading to these two paragraphs.

Please be accurate in your conclusions: “In our cohort, HTGAP more frequently had local complications compared with non-HTGAP and, consecutively, more severe forms of AP.” – only MSAP was significantly more common in HTGAP in your study.

We had more severe forms in HTGAP, you point it out right: statistically significant more MSAP, but also more SAP.

Minor issues:

The following sentences in the abstract should be rephrased to be more easily understandable: ”Male gender was predominant in the HTGAP group, at 76% (23/30) vs. 54% female 22 (126/232), p = 0.02; 53% (16/30) presented with DM vs. 18% who did not (42/232), p < 0.0001. The 23 patients with HTG presented higher CRP 48 h after admission: 207 mg/dl ± 3 mg/dl vs. 103 mg/dl ± 24 107 mg/dl, p < 0.0001.”

According to your suggestion, we rephrased these sentences.

Line 105: I suggest to cancel the sentence “Based on the recorded data for the NLR, the PLR was calculated at admission and at 105 48 h, using the international criteria (NLR = neutrophil count/lymphocyte count; PLR = platelet count/lymphocyte count).”. One cannot calculate PLR based on NLR.

 We modified this sentence.

Methods: “The patients were followed up for 90 days after discharge, with visits to the outpatient clinic or by phone.” – I wonder if this was also done retrospectively? (This would be suggested by the date of Ethical Committee approval).

Follow up is a general practice for these patients so it was done for the majority of our patients. 

Statistical analysis: The Authors state in Methods that median (range) is presented for non-normally distributed variables, however, I only see means and SD in Results (while I am sure that some variables, like e.g. CRP, are non-normally distributed in the studied groups).

Thank you, we corrected.

Line 134: “14% (39/262) had no etiology” – it is better to say that the etiology was unknown.

very good suggestion, we are not native speakers, and even if the paper was under language revision at mdpi words can be improved.

Figure 1. The data are fine, however, the flow chart must be arranged more aesthetically – its visual quality is very low.

We try to create and add a new flowchart (Figure 1). hope this is better!

Round 2

Reviewer 3 Report

The authors introduced many of my former comments and I think the results are now better presented. However, I still insist on correcting several minor issues.

The following sentence in the abstract still needs rephrasing, but it may possibly be done during the final manuscript edition. “HTGAP group differed from the nonHTGAP group in: the mean age 44.4 ± 9.2 vs 58.2 ± 17.1, p < 0.0001; , male gender 76% (23/30) vs. 54% male (126/232), p = 0.02; 53% (16/30) presented with DM vs. 18% (42/232), p < 0.0001,higher CRP at 48 h after admission: 207 mg/dl ± 3 mg/dl vs. 103 mg/dl ± 107 mg/dl, p < 0.0001”

Line 117: the units should be corrected (cell number/µL).

Line 128: The formula for BMI is incorrect.

Lines 128 and 130: there are no “international criteria” to calculate BMI, PLR, NLR; these are simply mathematical formulas (widely accepted).

Figure 1: Please follow the standard format of the patients selection diagram. The first version of the figure looked better, it only needed a more aesthetic arrangement: a consistent size of the font used, the size of the text boxes appropriate to the size of the font, and an ordered alignment of the text boxes.

Lines 192-193: Looking at the mean and SD, the length of hospital stay is surely non-normally distributed variable.

Table 1: In “Statistical analysis” authors state that for non-normally distributed variables, they report median and range while in Table 1 I see median and (unexpectedly) 95% confidence interval (for the median?). Please present data as median and range or median and lower-upper quartile.

Figures 2 and 3 should be named “Receiver operating characteristic curves for the five best predictors of SAP ….”

Table 3 and the related text (line 267): please clearly state if these date regard HTGAP.

Table 5: the specificity of BISAP is lacking.

Tables 2-5 should be commented in the main text of the results. Please add a single paragraph commentary regarding the similarities and differences between the predictors of severity in HTGAP and non-HTGAP.

Author Response

  1. The following sentence in the abstract still needs rephrasing, but it may possibly be done during the final manuscript edition. “HTGAP group differed from the nonHTGAP group in: the mean age 44.4 ± 9.2 vs 58.2 ± 17.1, p < 0.0001; , male gender 76% (23/30) vs. 54% male (126/232), p = 0.02; 53% (16/30) presented with DM vs. 18% (42/232), p < 0.0001,higher CRP at 48 h after admission: 207 mg/dl ± 3 mg/dl vs. 103 mg/dl ± 107 mg/dl, p < 0.0001”

We had the manuscript reviewed by Diagnostics English language services before submission, we are not native English language speakers. We rewrote this part of the abstract according to their recommendations. We dont know what else we could do, but we are willing to succed. 

2. Line 117: the units should be corrected (cell number/µL)

resolved

3. Line 128: The formula for BMI is incorrect.

resolved

4. Lines 128 and 130: there are no “international criteria” to calculate BMI, PLR, NLR; these are simply mathematical formulas (widely accepted).

resolved according to suggestions

5. Figure 1: Please follow the standard format of the patients selection diagram. The first version of the figure looked better, it only needed a more aesthetic arrangement: a consistent size of the font used, the size of the text boxes appropriate to the size of the font, and an ordered alignment of the text boxes.

we tried our best and reconfigure the first flow-chart according to suggestion

6.  Lines 192-193: Looking at the mean and SD, the length of hospital stay is surely non-normally distributed variable.

resolved as suggested

7. Table 1: In “Statistical analysis” authors state that for non-normally distributed variables, they report median and range while in Table 1 I see median and (unexpectedly) 95% confidence interval (for the median?). Please present data as median and range or median and lower-upper quartile.

resolved as suggested

8. Figures 2 and 3 should be named “Receiver operating characteristic curves for the five best predictors of SAP ….”

resolved as suggested

9. Table 3 and the related text (line 267): please clearly state if these date regard HTGAP.

resolved as suggested 

10. Table 5: the specificity of BISAP is lacking.

thank you, inserted in the table

11. Tables 2-5 should be commented in the main text of the results. Please add a single paragraph commentary regarding the similarities and differences between the predictors of severity in HTGAP and non-HTGAP.

inserted between lines 286-290, we focused on the good AUROC predictors and were different, hope this is fine with you, if not we are waiting for suggestions.

Thank you for your intense work in helping us to improve our paper!

the Authors